# Sequential Nitrification—Autotrophic Denitrification Using Sulfur as an Electron Donor and Chilean Zeolite as Microbial Support

**A. Barahona [1],\*, J. Rubio [1], R. Gómez [1], C. Huiliñir [2], R. Borja [3] and L. Guerrero [1]**

1 Laboratory of Wastewater Treatment, Department of Chemical and Environmental Engineering, Universidad Técnica Federico Santa María, Avenida España 1680, P.O. Box 110, Valparaíso 2340000, Chile
2 Laboratorio de Biotecnología Ambiental (LABIOTAM), Department of Chemical Engineering, Universidad de Santiago de Chile, Avenida Libertador Bernardo O'Higgins 3363, Santiago de Chile 9160000, Chile
3 Instituto de La Grasa (CSIC), Campus Universitario Pablo de Olavide, Edificio 46, Carretera de Utrera, Km 1, 41013 Sevilla, Spain
\* Correspondence: andrea.barahona@usm.cl

**Abstract:** Sequential nitrification–autotrophic denitrification (SNaD) was carried out for ammonium removal in synthetic wastewater (SWW) using sulfur as an electron donor in denitrification. Four reactors were operated in batch mode, two with zeolite (1 mm size) used as microbial support and two without support, to assess the effect of the zeolite addition in the SNaD. Aeration, anoxic, and anaerobic cycles were established, where 96% removal of $NH_4^+$-N (oxidized to nitrite or nitrate) was achieved in nitrification, along with 93% removal of $NO_3^-$-N in denitrification for the SNaD with zeolite. It was observed that the use of zeolite assists in buffering reactor load changes. Inhibition caused by nitrite accumulation in the denitrification stage was minimized by increasing the nitrogen concentration in the SWW. The results obtained in this study are the basis for the development of ammonium removal by simultaneous nitrification–autotrophic denitrification using a single reactor.

**Keywords:** batch reactor; nitrogen removal biotechnology; sequential nitrification-autotrophic denitrification; sulfur; zeolite

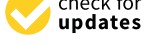



## 1. Introduction

The dumping of liquid waste in industries such as the agricultural or textile industries, with nitrogen content in the form of ammonium ($NH_4^+$-N) in water sources, leads to eutrophication, acidity, and toxicity for the ecosystem [1]. At present, there are different biological processes for the removal of $NH_4^+$-N found in wastewater flows, particularly conventional nitrification–denitrification (ND), which has shown high levels of efficiency; however, in wastewaters with a low C/N ratio, additional organic matter (OM) must be added for heterotrophic microorganisms, which leads to higher operating costs. One solution to overcoming this disadvantage is to use anaerobic autotrophic microorganisms in denitrification [2,3], which implies the elimination of the additional organic substrate requirements and avoids potential secondary pollutants.

Autotrophic denitrification (aDN) uses hydrogen and sulfur compounds ($H_2S$, $S^0$, $S_2O_3^{2-}$, $S_4O_6^{2-}$, and $SO_3^{2-}$) as electron donors and inorganic carbon ($CO_2$, $HCO_3^{2-}$) as a carbon source for microorganism growth. In comparison to conventional heterotrophic denitrification (hDN), aDN has two clear advantages: (1) it does not need an external source of organic carbon, and (2) it reduces the production of biomass, thus minimizing sludge management [3–6].

The use of nitrification–autotrophic denitrification (NTaDN) has gained attention for years [1,7,8] due to the need to reduce the use of carbon as an electron donor in order to minimize the risks associated with induced contamination (as in the case of hDN)

based on the addition of organic matter in processes prior to discharge from treatment plants. Currently, there are factories in the agricultural and swine industries that dispose of wastewaters with high nitrogen and sulfur compound concentrations, or others such as leather and fertilizer processing and leachate lakes that yield wastewater with a low C/N [9]. The different studies have focused on how to jointly remove nitrogen and sulfur from these wastewaters [7,10,11], while at the same time eliminating the risk associated with the presence of waste compounds in the treatment plants' effluents caused by added organic matter, as in the case of hDN [6,12]. Moreover, some features of the autotrophic and heterotrophic processes may complement each other, such as reducing the demand for organic carbon and basicity. This makes it possible to combine the two processes in order to overcome their respective defects, and this combination may be used for the abovementioned wastewaters to simultaneously remove carbon, nitrogen, and sulfur.

Nitrification is a mechanism by which autotrophic microorganisms obtain energy for their metabolism from the oxidation of reduced nitrogen species (e.g., $NH_4^+$), using oxygen as the oxidizing agent. On the other hand, denitrification is a mechanism for obtaining metabolic energy from the reduction of oxidized nitrogen species (e.g., $NO_3^-$) to $N_2$, using organic or inorganic compounds as reducing agent under anaerobic conditions. In this sense, heterotrophic denitrifying microorganisms oxidize organic matter and autotrophic denitrifying microorganisms oxidize inorganic compounds.

The joint removal of sulfur and nitrogen can help leverage the benefits of the interaction between sulfur and nitrogen cycles, where oxidized nitrogen compounds are reduced to $N_2$, while the reduced sulfur compounds are oxidized to sulfate ($SO_4^{2-}$), producing a low negative impact on the environment [7,13,14].

Reductive sulfur compound is one of the most common inorganic electron donors performing autotrophic denitrification. Up until now, a variety of sulfur species, including $S^{2-}$, $HS^-$, chemical and biogenic $S^0$ and $S_2O_3^{2-}$, have been successfully applied for sulfur-based autotrophic denitrification process [14]. When $S^{2-}$ is used as an electron donor to conduct denitrification, $S^{2-}$ first reacts with $NO_2^-$, then with $NO_3^-$ as a result, and competition for the electron acceptor of $NO_2^-$-N can occur during the coupling of $S^{2-}$-denitrification, affecting the nitrogen removal efficiency of the reactor [15]. Further, the $S^0$ produced by the process is difficult to separate from the reactor, and excessive $S^0$ accumulated on the surface of microorganisms will affect the long-term stable operation of the reactor [15]. For that reason, an appropriate rate S/N to obtain optimum operation, and the definition of conditions of process are necessary where this work is part of a project research.

The characteristics and selection of appropriate mediums for microbial cultures have been shown as important for improving the operating conditions in the mixed liquid. For inoculums with slow growth rates, as in the case of the nitrifying inoculum, reactors with biofilm can be of great use [16,17]. Montalvo et al. [18] proposed that these mediums must meet the following characteristics: the surface must favor the colonization of the microorganism; it must be physically and chemically resistant; and it must be relatively inert.

Different studies [19–23] have shown that the use of zeolite meets the characteristics described above and has already produced favorable results. Huiliñir et al. [21] demonstrated that the use of zeolite reduces the inhibition effects of organic matter and sulfur ($HS^-$) in the nitrification processes and increases the rate of removal of total ammonia and the rate of nitrate production in the same way as obtained by Montalvo et al. [23]. Cortés [19] obtained a higher index of microbial growth and cellular retention time in the mixed nitrifying–denitrifying culture when using zeolite as a medium in comparison to a control reactor that did not use it as such. Meanwhile, Mery [22] was able to obtain adhered microorganisms where only 0.6% of the biomass was suspended within the reactors. Montalvo et al. [7] managed to reduce the stabilization time of an autotrophic denitrifying inoculum by up to 50% when using zeolite as a medium.

Therefore, the objective of this study was to assess the operation of sequential batch reactors for the nitrification–autotrophic denitrification using an enriched mixed inoculum

for the elimination of $NH_4^+$-N with $S^0$ as an electron donor in denitrification, Chilean zeolite as a microbial medium, and synthetic wastewater (SWW) as a substrate. This is framed within a larger study and seen as the first step towards simultaneous nitrification-autotrophic denitrification using a single reactor, which guarantees the novelty of this study.

## 2. Materials and Methods

### 2.1. Enrichment of Microorganisms in aDN

This assay was performed in a batch reactor with an effective volume of 1.5 L, operated at 25 °C, a pH of 8, and a nitrogen loading rate (NLR) of 0.061 kg N/(m$^3$·d). The reactor had a two-blade stainless steel mixer, whose axis was covered with a sleeve to create a water seal to minimize the entry of air into the reactor. To obtain the bacteria-enriched inoculum with tolerance to ammonia and sulfur compounds, the sludge used was taken from the wastewater treatment plant of a swine production company. The reactor was inoculated with 0.225 L of sludge, which presented a concentration of 85 g VSS/L. It also presented a high concentration of ammonium, which validates that the microorganisms found in the sludge were resistant to the presence of nitrogen compounds.

The SWW used as the substrate for the enrichment of the aDN was prepared according to the specifications reported by Koenig and Liu [24] with slight modifications, according to the composition detailed in Table 1. In this case, elemental sulfur was added and sifted to 500 µm at a ratio of 0.35 g $S^0$/g $KNO_3$. The substrate was diluted to 1 L with distilled water. The equation for aDN is shown in Equation (1) below.

$$6NO_3^- + 5S^0 + 2H_2O \rightarrow 3N_2 + 5SO_4^{-2} + 4H^+ \tag{1}$$

**Table 1.** (a) Composition of synthetic wastewater used in the autotrophic denitrification, (b) composition of micronutrients.

| Component | Unit | Quantity | Component | Unit | Quantity |
|---|---|---|---|---|---|
| $K_2HPO_4$ | g/L | 2.00 | $Na_2MoO_4$*$7H_2O$ | g/L | 1.00 |
| $NaHCO_3$ | g/L | 1.00 | $FeSO_4$*$7H_2O$ | g/L | 30.00 |
| $KNO_3$ | g/L | 2.00 | $ZnCl_2$ $4H_2O$ | g/L | 1.00 |
| $NH_4Cl$ | g/L | 0.50 | $CaCO_3$ | g/L | 2.00 |
| $MgCl·6H_2O$ | g/L | 0.50 | $MnCl_2$ $4H_2O$ | g/L | 1.50 |
| $S^0$ * | g/L | 0.70 ** | $CuSO_4$ $5H_2O$ | g/L | 0.25 |
| Micronutrients *** | mL/L | 0.50 | $CoCl_2$ $6H_2O$ | g/L | 0.25 |
| | | | HCl | g/L | 50.00 |
| | | | $NiCl_2$ $6H_2O$ | g/L | 0.25 |
| | | | $H_3BO_3$ | g/L | 0.50 |
| | (a) | | | (b) | |

Note(s): * Average particle size (dp < 500 µm); ** amount calculated for the autotrophic denitrification reaction using elemental sulfur according to Campos et al. [5]; *** according to Fajardo et al. [25].

### 2.2. Validation of Nitrifying (NT) Microorganisms

The nitrification inoculum was obtained from the study by Gómez [26], whose VSS (volatile suspended solids) concentration was 19.7 g VSS/L; that study performed simultaneous nitrification–autotrophic denitrification using sodium thiosulfate as an electron donor for aDN. The mixed inoculum presented a removal efficiency of 98% in terms of ammonium oxidation and nitrate reduction, as well as conditions for nitrification with the presence of sulfur compounds (sulfates). The validation of the inoculum was performed in a batch reactor with an effective volume of 1.5 L, operated at 25 °C, a pH of 7.5, a dissolved oxygen (DO) concentration between 3 and 5 ppm, and an NLR of 0.161 kg N/(m$^3$·d). The reactor had a two-blade stainless steel mixer, in addition to four diffusers located equidistantly at the base of the reactor, which allowed for the entry of air in the form of microbubbles.

The reaction governing nitrification is described by Equations (2) and (3) below:

$$2NH_4^+ + 3O_2 \rightarrow 2NO_2^- + 4H^+ + 2H_2O \tag{2}$$

$$2NO_2^- + O_2 \rightarrow 2NO_3^- \tag{3}$$

Six nitrification cycles were performed to validate the efficiency of the inoculum. Using the SWW specified in the study by Beristain-Cardoso et al. [27] with slight modifications and described in Table 2, it was then aerated for 8 h to evaluate the removal of $NH_4^+$.

**Table 2.** (a) Composition of synthetic wastewater used in nitrification, (b) composition of micronutrients.

| Component | Unit | Quantity | Component | Unit | Quantity |
|---|---|---|---|---|---|
| $K_2HPO_4$ | g/L | 3.5 | EDTA | g/L | 5.0 |
| $KH_2PO_4$ | g/L | 4.0 | $CuSO_4\ 5H_2O$ | g/L | 1.57 |
| $NaHCO_3$ | g/L | 2.4 | $CaCl_2\ 2H_2O$ | g/L | 5.54 |
| $NH_4Cl$ | g/L | 0.5 | $MnCl_2$ | g/L | 5.0 |
| NaCl | g/L | 0.2 | $(NH_4)_6Mo_7O_{24}\ 4H_2O$ | g/L | 1.1 |
| Micronutrients | mL | 0.5 | $FeCl_3$ | g/L | 5.0 |
| | | | $CoCl_2\ 6H_2O$ | g/L | 1.6 |
| | | | $MgCl_2\ 5H_2O$ | g/L | 5.0 |
| (a) | | | (b) | | |

### 2.3. Zeolite

Different previous studies [18–22,28] have shown that the zeolite to be used should meet the following characteristics: the surface must favor the colonization of the microorganism; it must be physically and chemically resistant; and it must be relatively inert. Huiliñir et al. [21] demonstrated that the use of zeolite reduces the inhibition effects of organic matter and sulfur ($HS^-$) in the nitrification processes and increases the rate of removal of total ammonia and the rate of nitrate production in the same way as obtained by Montalvo et al. [23].

### 2.4. Sequential NTaDN Process

This study used four sequential batch reactors, all with the same characteristics, including an amber glass structure, a diameter of 0.12 m, a height of 0.18 m, and an effective volume of 1.5 L. Two of the four reactors were used for the nitrification phase and had four diffusers located equidistant at the base of the reactor, which allowed for the entry of air in the form of microbubbles. With respect to the reactors used for aDN, the only difference was that the air diffusers were not installed in these ones. The four reactors had a two-blade stainless steel mixer, whose axis was covered with a sleeve to create a water seal to minimize the entry of air into reactor.

A set of four reactors (two reactors for NT and other two for aDN) were used. Two reactors were loaded with a microbial medium containing 1 mm zeolite, with a concentration of 20 g zeolite/gVSS [5]. These reactors were labelled as R1 NT and R1; the configuration of SBR reactors is showed in Figure 1. The remaining reactors did not receive zeolite in order to validate the performance of the medium in the individual process, and these were labelled as R0 NT and R0 aDN.

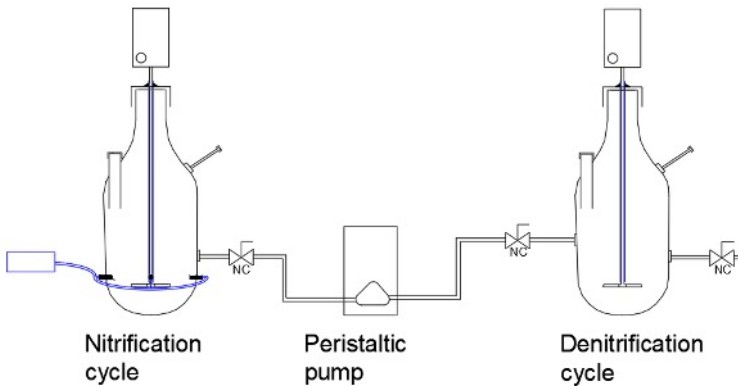

**Figure 1.** Configuration of SBR reactors. Sequential process circuits constructed: Nitrification (NT) reactor and Autotrophic denitrification (aDN) reactor.

### 2.4.1. Inoculum for the Sequential Process

The two reactors with the aeration system were loaded with nitrifying inoculum, occupying 20% of the effective volume of the reactor (0.3 L of inoculum with 14.7 g VSS/L achieving 2940 mg VSS in liquid/L in reactor). Likewise, the reactors without the aeration system were loaded with denitrifying inoculum, also occupying 20% of the effective volume of the reactor (0.225 L of inoculum with 17.6 g VSS/L achieving 2992 mg VSS in liquid/L in reactor).

### 2.4.2. Microbial Medium

The concentration of zeolite loaded in the R1 reactors was 20 g zeolite/g VSS [5]. The zeolite used was previously sifted by size and classified by the granulometry analysis of coarse and fine aggregates (ASTM C-136-01) to obtain a diameter of 0.9–1.12 mm. The chemical composition of the zeolite used in the study was $SiO_2$, 64.19%; $TiO_2$, 0.51%; $Al_2O_3$, 11.65%; $Fe_2O_3$, 2.53%; $MnO$, 0.03%; $MgO$, 0.66%; $CaO$, 3.42%; $Na_2O$, 0.75%; $K_2O$, 1.60%; $P_2O_5$, 0.03%; and $P_xC$, 14.64% (calcination lost).

### 2.4.3. Operation Strategy

As mentioned in Section 2.4, two sequential process circuits were constructed and composed of one NT reactor and one aDN reactor (Figure 1). The composition of synthetic wastewater in sequential aDN process is presented in Table 3, and times for each phase of the SBR process are detailed in Table 4.

**Table 3.** Composition of synthetic wastewater in sequential aDN process.

| Component | Unit | Quantity |
|---|---|---|
| $K_2HPO_4$ | g/L | 2.00 |
| $NaHCO_3$ | g/L | 1.00 |
| $MgCl\ 6H_2O$ | g/L | 0.50 |
| $S^o$ * | g/L | 0.33 ** |
| Micronutrients *** | mL | 0.50 |

Note(s): * Average particle size (dp < 500 μm) (Hashimoto et al. [29]; Kuai and Verstraete [30]); ** amount calculated for the autotrophic denitrification reaction using elemental sulfur according to Campos et al. [5] assuming complete nitrification; *** according to Fajardo et al. [25] (Table 1b).

The SBR operating phases include fill time, reaction time, sedimentation time, purge time, and inactive time. These phases apply to each reactor (NT and aDN), with the difference being that the reaction phase of the NT reactor is comprised of an aerobic cycle followed by an anoxic cycle (to reduce DO), while the reaction phase of the aDN reactor is comprised of an anaerobic cycle (in the absence of $O_2$).

**Table 4.** Operating conditions for the sequential NTaDN process.

| Parameter | R NT | R aDN |
|---|---|---|
| Temperature | 25 °C in liquid, 31 °C in chamber | 25 °C in liquid, 31 °C in chamber |
| Feed pH | 7.0–8.0 | 7.0–8.0 |
| Medium (zeolite) | according to reactor | according to reactor |
| RPM | 120 RPM | 120 RPM |
| $T_{Ll}$ | 0.02 h | 0.02 h |
| $T_R$ | 20 h | 24 h |
| Aerobic cycle | 8 h | - |
| Anoxic cycle | 12 h | - |
| Anaerobic cycle | - | 24 h |
| $T_{Ssed}$ | 0.5 h | 0.5 h |
| $T_{Dec}$ | 0.03 h | 0.03 h |
| $T_{In}$ | 0.5 h | 0.5 h |
| CRT | 26.3 d | 31.5 d |
| HRT | 2.84 d | |
| NLR | 0.046 kg N/m$^3$ d | |

Note(s): $T_{Ll}$: Fill time; $T_R$: Reaction time (aerobic cycle + anoxic cycle + anaerobic cycle); $T_{Sed}$: Sedimentation time; $T_{Dec}$: Decant time; $T_{In}$: Inactive time. CRT: Cellular retention time. HRT: Hydraulic retention time. NLR: Nitrogen loading rate.

The aerobic cycle was set at 8 h, the anoxic cycle at 12 h (to reduce DO), and the anaerobic cycle (in the absence of O$_2$) at 24 h. All of these times were preselected based on the results obtained by Gómez [26]. After the aerobic cycle, the accumulated oxygen was left to be consumed by the autotrophic nitrifying microorganisms in the nitrifying reactors (with the aeration system). Once the concentration dropped to 0.5 mg DO/L, a peristaltic pump was used to transfer the SWW from the nitrifying reactor to the denitrifying reactor.

### 2.4.4. Substrate

Throughout the sequential process the composition of SWW described in Table 2 was used, adding the compounds described in Table 3.

The effluent from the nitrification reactor is transferred to the denitrification reactor as influent for the denitrification process.

### 2.4.5. Operating Conditions and Measurement Variables

The operating conditions for the sequential process are presented in Table 4.

At the beginning of the NT, the nitrogen compounds (ammonium, nitrite, and nitrate) were measured, along with the pH, dissolved oxygen (DO), and the airflow using the flowmeter. For the purposes of this study, the NT ended once the ammonium oxidation was greater than 95%. Measurement was taken every hour. Once the NT was complete, the specific DO measurements were taken every hour to validate its reduction and to complete the anoxic phase. Then, the nitrogen compounds (ammonium, nitrite, and nitrate) were measured, in addition to the oxidized form of sulfur (sulfate) at the beginning and end of the aDN. For the purposes of this study, the aDN ended once the nitrate reduction was greater than 95%. During the research study, it was assumed that the nitrogen missing to complete the matter balance was converted to N$_2$ at the end of the aDN. All the analyses were performed in triplicate and they were carried out in accordance with standard methods [31].

### 2.4.6. Operation of the Sequential aDN Process

To evaluate the resistance of each inoculum to changes in the influent composition, an increase in $NH_4^+$-N composition by 15% was established with respect to the previous period.

## 3. Results and Discussion

### 3.1. Enrichment of aDN Inoculum

The anaerobic process was monitored for 67 days to obtain a microbial culture that could reduce nitrate using elemental sulfur as electron donor. The results obtained during this period are shown in Figure 2. The first two cycles of aDN were unable to remove all nitrate in the SWW, where each cycle represents a period of time. For the first cycle, 69.2% of $NO_3^-$-N was removed during an 11-day period. For the second cycle, the $NO_3^-$-N removal was 88%, with an increase in the removal rate from 23.16 mg $NO_3^-$-N/(L·d) to 42.15 mg $NO_3^-$-N/(L·d). For the third and fourth cycles, the removals were 93.6% and 96.3%, respectively, maintaining a removal rate of 42.15 mg $NO_3^-$-N/(L d). For this enrichment phase, the highest removal rate was registered on day 48 at 76.9 mg $NO_3^-$-N/(L·d).

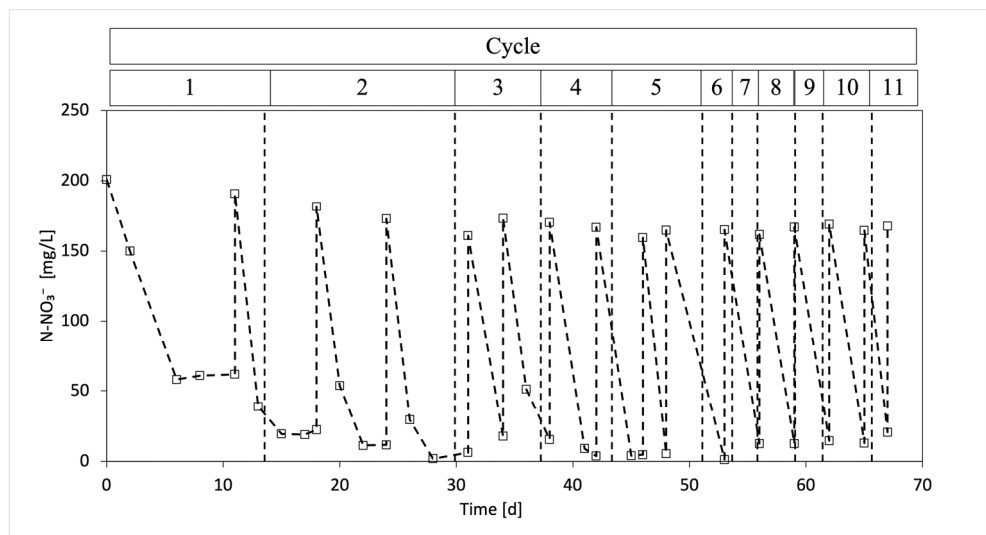

**Figure 2.** Variation of the concentration of N-$NO_3^-$ over time during the enrichment of the denitrifying inoculum.

With respect to the sulfate balance produced, Figure 3 shows the variation of sulfate concentration over time during this denitrifying step (aDN enrichment). As for the fourth cycle (day 26), the production of sulfates reflected the removal of nitrates in the reactor. It was assumed that the removal of nitrate during the initial cycles (1st to 3rd cycle) reflected heterotrophic denitrification processes where the microorganisms used the organic matter for nitrate reduction. It was not until the fourth cycle that the formation of sulfates corresponded to nitrate consumption (2.51 g S-$SO_4^{2-}$/$NO_3^-$-N) according to the reaction of aDN with $S^0$ as reported by Campos et al. [5]. The variation of VSS over time in enrichment process is in the Supplementary Material (Figure S3).

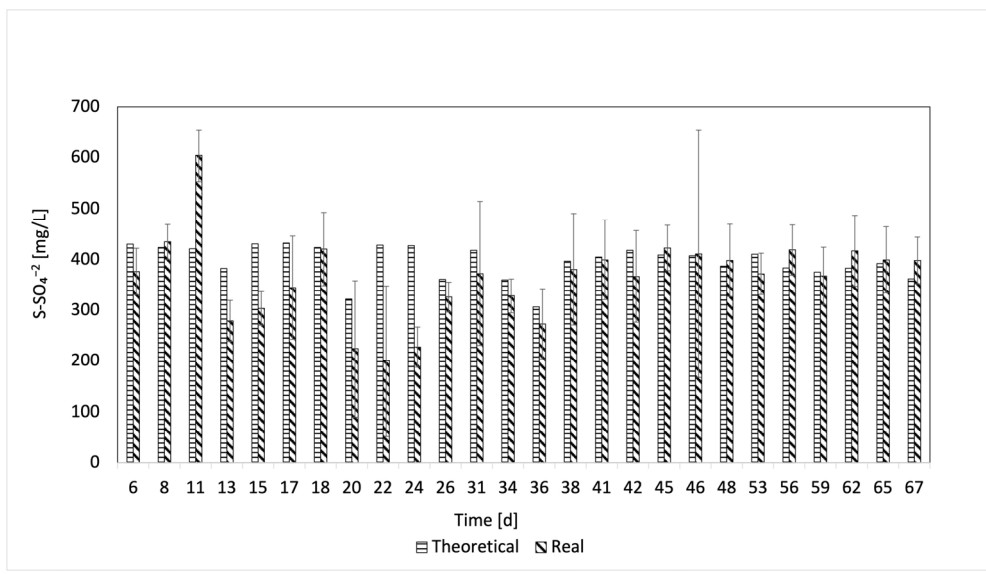

**Figure 3.** Sulfate balance in the second Autotrophic denitrification (aDN) enrichment.

### *3.2. Validation of NT Inoculum*

The results obtained during the validation cycles for the nitrifying inoculum showed total nitrification, where there was no accumulation of nitrite during each 8-h cycle, and the transformation of ammonium to nitrate was greater than 99%. From the balance performed (Figure 4), it was observed that, on days 3 and 5, more nitrogen was obtained at the end of the process, which occurred because there was nitrate from the previous cycle within the accumulated volume (500 mL). To avoid this accumulation, the biomass was washed at the beginning of each cycle. For the later cycles (4, 5 and 6), the nitrogen balance was closed with the addition of the buffer solvent (monopotassium and dipotassium phosphate) to the SWW, while the pH was maintained during the process at values of 7.5–7.8. The enriched aerobic inoculum showed a brown color in comparison to the enriched anaerobic inoculum, which was black.

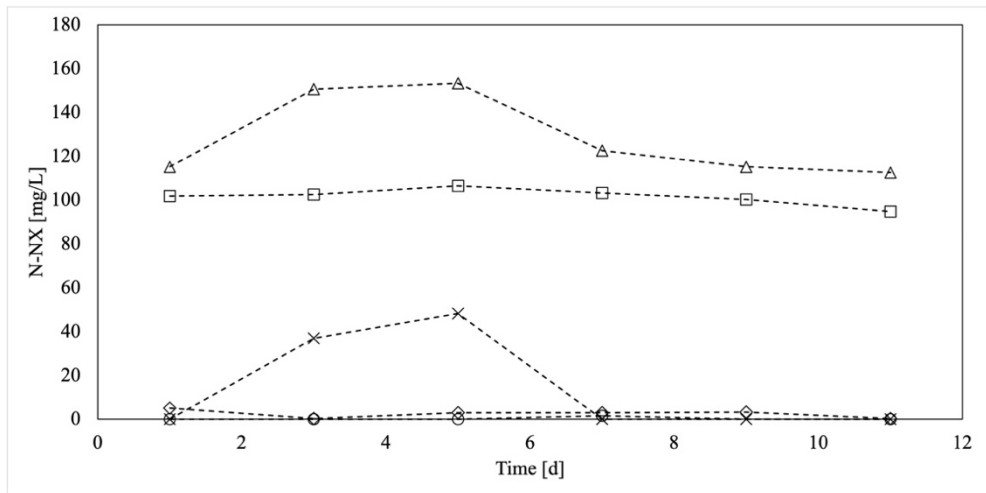

**Figure 4.** Evaluation of the Nitrification (NT) inoculum during the enrichment phase. N-NH$_4^+$ influent ($\square$), N-NH$_4^+$ effluent ($\lozenge$), N-NO$_2^-$ effluent ($\bigcirc$), N-NO$_3^-$ influent ($\times$), and N-NO$_3^-$ effluent ($\triangle$).

The objective of the nitrifying inoculum was to evaluate the process performance and not to enrich, since the inoculum was already enriched from the work of Gómez [26]. By measuring the VSS at the beginning and end of the validation period, a 15% reduction

in biomass was observed, which could be associated with loss during the sampling and adherence to the air diffusers in the reactor.

### 3.3. Operation of the Sequential aDN Process

According to the results obtained, the cycles established in Section 2.4.3 were modified to 5 h for the aerobic cycle, 4 h for the anoxic cycle, and 15 h for the anaerobic cycle.

### 3.3.1. Effect on Nitrification with or without Zeolite

The reactors in the NT phase were operated in the reaction phase, maintaining 3–5 mg $O_2$/L and 0.6–1.0 L air/h. In the evaluation of this step, it was shown that the enriched inoculum was capable of oxidizing all ammonium when fed at a concentration of 131 mg N/L, maintaining an average biomass concentration of 3500 mg VSS/L; the variation of VSS over time in sequential aDN process appears in the Supplementary Material (Figure S4).

Profile of sequential process with and without zeolite during reactor stabilization is presented in the Supplementary Materials (Figures S5–S8).

Four operation periods were performed, with changes in nitrogen concentration in the form of $NH_4Cl$, which was increased by 15% between periods. The first period was performed with a feed of 0.4 g $NH_4Cl$/L, corresponding to 105 mg N/L. Once the aeration time was complete, the anoxic phase began, maintaining agitation for the nitrifying microorganisms to reduce the oxygen concentration in the reactor prior to the aDN phase. The anoxic phase was defined at 4 h, the time needed to reduce the concentration of DO < 0.5 mg $O_2$/L.

The overall results obtained are presented in Figures 5–8, which detail the nitrogen concentration in the different compounds of it found in the reactor (Figures S1 and S2 see Supplementary Material); for the beginning of the cycle (I), end of the nitrification phase and beginning of denitrification (N), and end of denitrification (D). Moreover, stages conditions are: stage 1: NRL 0.066 kg N/(m³·d), 1–10 d; stage 2: (NRL 0.076 kg N/(m³·d), 10–19 d; stage 3: NRL 0.082 kg N/(m³·d), 19–28 d); stage 4: NRL 0.094 kg N/(m³·d), 28–37 d. Figures 6 and 8 show the percentage of removal during each cycle evaluated.

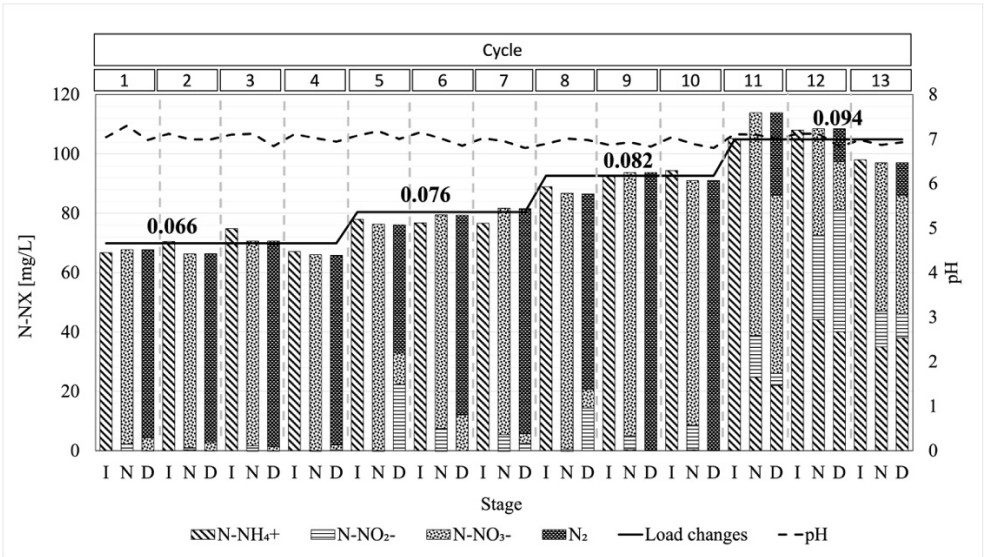

**Figure 5.** Results of the sequential aDN process. Nitrogen balance R0. Beginning of the cycle (I), end of the nitrification phase and beginning of denitrification (N), and end of denitrification (D). Numbers above black line represent NRL: kg N/(m³·d).

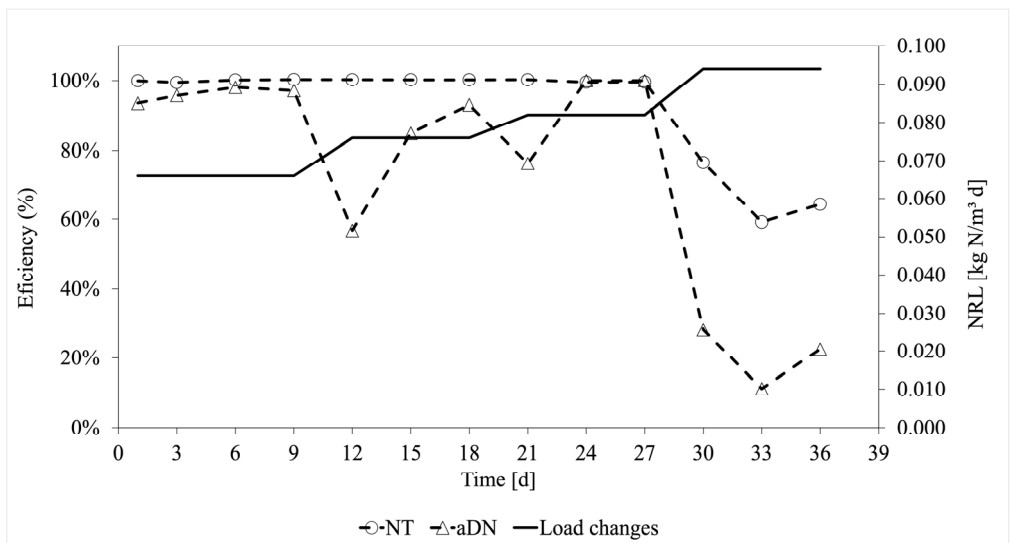

**Figure 6.** Results of the sequential aDN process. Nitrification (NT) and Autotrophic denitrification (aDN) efficiencies in R0.

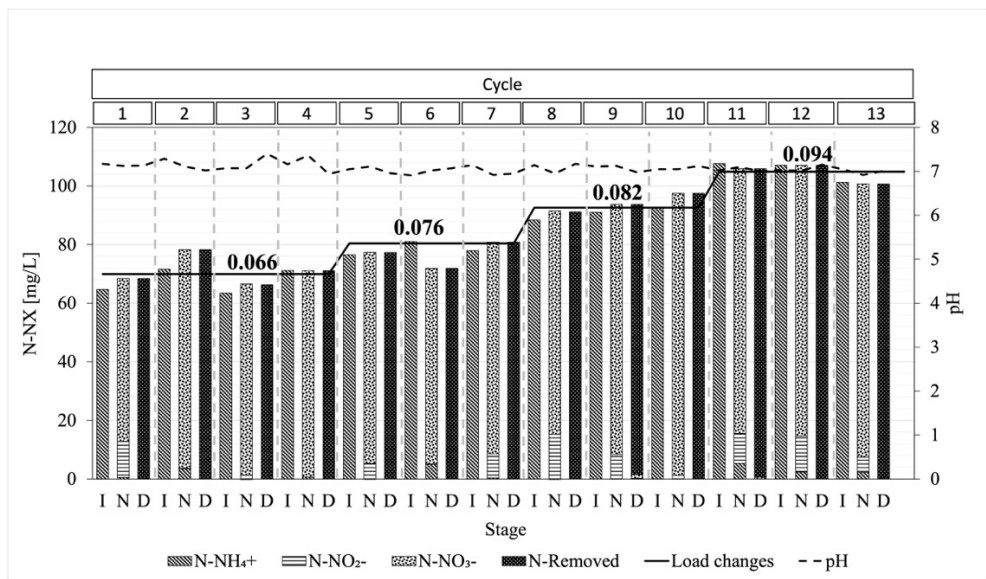

**Figure 7.** Results of the sequential aDN process. Nitrogen balance R1. Beginning of the cycle (I), end of the nitrification phase and beginning of denitrification (N), and end of denitrification (D). Numbers above black line represent NRL: kg N/(m$^3$·d).

Based on the results obtained from the sequential process without zeolite, it was noted that the R0-NT reactor presented complete removal of $NH_4^+$-N with an average of 99.69% up to the third cycle, whose NLR was 0.082 kg N/(m$^3$·d). This result was consistent with the results obtained by Gómez [26] for the reactor without zeolite, except that in the latter study, the reactor had a concentration of 5000 mg VSS/L and an aeration time of 16 h, which was much higher than that used in the present research study (5 h). The difference in time is due primarily to the absence of agitation, especially during the anoxic phase, when the removal of $NH_4^+$-N was still performed by the residual dissolved oxygen in the aerobic phase.

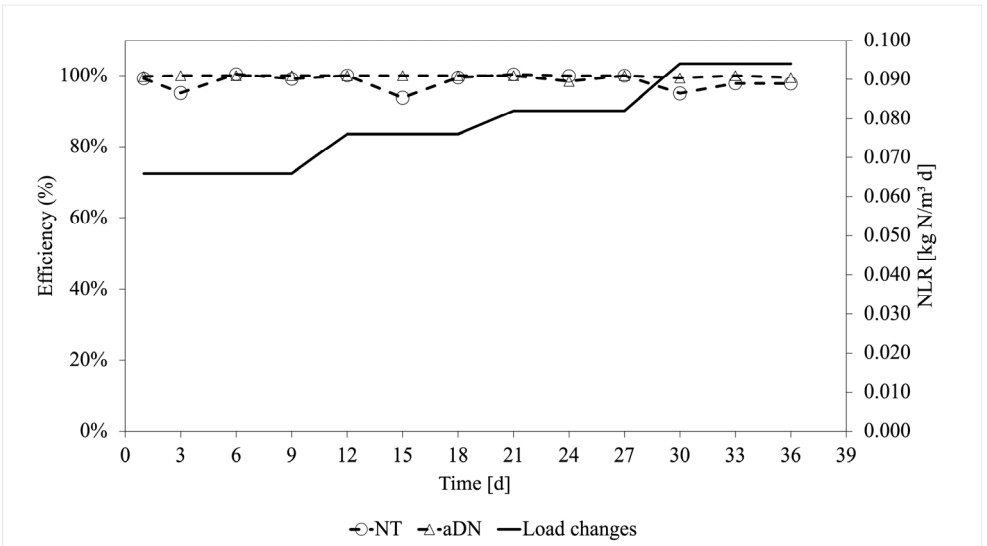

**Figure 8.** Results of the sequential aDN process. Nitrification (NT) and Autotrophic denitrification (aDN) efficiencies in R1.

For the fourth cycle, whose NLR was 0.094 kg N/(m$^3$·d), the removal dropped to 67%. As mentioned in the methodology, a buffer solvent was added to the SWW in the feed to help maintain the pH between 7.0 and 8.0 during the process phases. For this reason, inhibition due to the presence of free ammonium was discarded since this is generally found at a pH 9 or NH$_3^-$-N concentration higher than 7 mg/L [32]. Therefore, the drop in efficiency was attributed to the increase in nitrogen load in the reactor, resulting in an incomplete nitrification and the accumulation of nitrite in the SWW (41 mg NO$_2^-$-N/L). There was inhibition in the aDN phase for the fourth period, where nitrate removal dropped to 21%.

With respect to the nitrification phase in the reactor with zeolite (R1-NT), the removal efficiency was over 95% during the periods performed in the present study. The reactor supported up to an NLR of 0.094 kg N/(m$^3$·d) and maintained an average removal efficiency of NH$_4^+$-N at 98.87% during the first three cycles. This efficiency was slightly lower than that achieved in the reactor without zeolite because the efficiency in cycles 2 and 6 was lower at the beginning of each period but recovered during the following cycle. For the last period evaluated (cycles 11, 12, and 13), the efficiency was 95.1%, 97.2%, and 98.1%, respectively. With respect to the pH in the reactor, it was maintained at an average value of 7.0 (see Figure 7). At the beginning of each period, there was a small accumulation of nitrite (under 20 mg N/L), which dropped as each new cycle began.

During the NT process, the use of zeolite favors the inoculum's quick adaptation to changes in NH$_4^+$-N concentration in the feed, because the zeolite, in addition to operating as a microbial support medium, has ionic exchange properties [33], where part of the ammonium input can be absorbed on the zeolite surface, which acts as a concentration regulator within the mixed liquid, and is later biologically metabolized by the nitrifying microorganisms [33]. Similar results were reported by Montalvo et al. [18], Cortés [19], González [20], Huiliñir et al. [21], Mery [22], and Gómez [26], where their reported processes presented better results in the nitrifying activity than in the processes that had no zeolite as a microbial support medium. In fact, in this research, zeolite decreased 70% of the time in nitrification (data not shown) in concordance with Cortés [19] and Gómez [26].

At the end of each phase, the VSS measured in each reactor did not show any significant difference in the change in biomass concentration inside the reactor, showing an average of 3500 ± 200 mg VSS/L for the two reactors (Figure S4, Supplementary Material).

### 3.3.2. Effect on Denitrification with or without Zeolite

In terms of the aDN, it was monitored in the first cycle of the sequential process to determine the anaerobic operation time in the aDN, which was used across all cycles. It was observed that, after 15 h, there was a nitrate removal of over 95% in the first cycle, and this time was used as a reference for nitrate removal.

The aDN, the process without zeolite, showed great variability to changes in nitrogen concentration. For instance, for the first cycle of the second period (cycle 5, NLR 0.076 kg N/(m$^3$·d)), the removal in the aDN dropped to 57% in comparison to the initial period, where the average removal was 96.12%; however, for the following cycles (6 and 7), the removal efficiency in the aDN increased to 85% and 95%, respectively. The same occurred at the beginning of the third period of aDN, when the NLR increased to 0.082 kg N/(m$^3$·d), the efficiency in the aDN dropped to 76%, but, in the later cycles (9 and 10), it recovered efficiencies of 99.2% and 98.5%, respectively. The results show the sensitivity of the aDN process to changes in nitrogen concentration in the feed. However, when maintaining operating conditions, the process can recover when maintaining low concentrations of possible inhibitor elements such as $NO_2^-$-N, OD ($O_2$), and a low pH (<6.5).

The opposite occurred in the fourth period, where the nitrite accumulation, due to incomplete nitrification in the first phase of the process without zeolite (Figures 5 and S1, Supplementary Material), generated inhibition in the reduction of nitrogen compounds during the aDN phase, achieving a maximum removal of 21%. Alzate et al. [28] reported the presence of inhibition in aDN processes when there is accumulation of $NO_2^-$ with concentrations of over 1 mg/L. However, cycles 5 and 8 of the process without zeolite showed that the inoculum recovered in cycles where the nitrite concentration was less than 23 mg $NO_2^-$-N/L. On the other hand, Fajardo et al. [25] reported process inhibition in the presence of nitrite over 35 mg $NO_2^-$-N/L; therefore, the inhibition in the aDN during the fourth period of the process without zeolite may be attributed to the presence of nitrite due to incomplete nitrification, since the mixed liquid had a concentration of 41 mg $NO_2^-$-N/L.

Regarding the aDN in the process with zeolite, it is clear (Figure 8) that the changes in nitrogen concentration in the feed did not affect the nitrate reduction, allowing for an average removal rate of 99.79%. Therefore, it can then be concluded that the enriched inoculum in the process with zeolite can perform the aDN without being subject to inhibition factors, contrary to what occurred in the process without this support medium.

In terms of the use of zeolite in the aDN phase, it was not possible to determine any contribution to process recovery after receiving increases in the NRL. For the two cases studied (with and without zeolite), the inoculum recovered in the following cycle and the removal capacity increased.

On the other hand, the importance of agitation for carrying out biological processes is clear, where aeration has the main objective of supplying oxygen to the aerobic cultures. However, in many cases (at industry scale, for example), aeration systems play a second role, which is to maintain agitation of the mixed liquid. This was the case reported by Gómez [26], where the aeration system also played the role of keeping the reactor liquid mixed. However, for the anoxic and anaerobic phases, the biological process had no system for mixing. For this reason, the anoxic and anaerobic aeration times were greater in the study by Gomez [26] in comparison to those determined in the present study.

### 4. Conclusions

A novelty sequential nitrification–autotrophic denitrification study has been performed using elemental sulfur as an electron donor and Chilean zeolite as a microorganisms' immobilization medium in comparison to another similar process without such medium.

The operation strategy of the sequential process must be improved with the aim of a simultaneous process, which is the objective of the overall project in which this study is framed; however, a breakthrough was achieved by using sulfur and ammonium for this study, where high removal efficiencies were achieved in each stage.

The process operated without zeolite generated complete nitrification and an average removal of $NH_4^+$-N of 99.69% until the third period, with an NLR of 0.082 kg N/($m^3 \cdot$d), and the nitrate removal during denitrification achieved a value of 95%. For the fourth period, where the NLR was increased to 0.094 kg N/($m^3 \cdot$d), the efficiency in nitrification decreased to 67%. As a result of incomplete nitrification and the accumulation of nitrite in the SWW (41 mg $NO_2^-$-N/L), there was inhibition in the aDN phase for the fourth period, where nitrate removal dropped to 21%.

With respect to the process with zeolite at an average size of 1 mm, the nitrification was complete for the four periods evaluated, with average removal of $NH_4^+$-N of 98.87%, at a NLR of 0.094 kg N/($m^3 \cdot$d), whereas the nitrate removal during denitrification reached 99.79%.

It can be stated that zeolite contributes to an operation with high efficiency at the same load charge (NLR of up to 0.094 kg N/($m^3 \cdot$d)); moreover, without inhibition factors, no accumulation of nitrite occurred during the process in contrast with operation without zeolite. In addition to all of the above, in the NT process, the use of zeolite favors the inoculum's quick adaptation to changes in $NH_4^+$-N concentration in the feed. This confirms that the contribution of zeolite to its operation as a microbial support medium has cationic exchange properties, where part of the ammonium fed can be absorbed on the zeolite surface, which acts as a concentration regulator within the mixed liquid, and is later biologically transformed by the nitrifying microorganisms.

**Supplementary Materials:** The following supporting information can be downloaded at: https://www.mdpi.com/article/10.3390/w15010095/s1. Figure S1. Results of the sequential aDN process. NLR R0, the beginning of the cycle (I), end of the nitrification phase and beginning of denitrification (N) and end of denitrification (D). Figure S2. Results of the sequential aDN process. NLR R1, the beginning of the cycle (I), end of the nitrification phase and beginning of denitrification (N) and end of denitrification (D). Figure S3. The variation of VSS over time in enrichment aDN process. Figure S4. The variation of VSS over time in sequential NT-aDN process. Figure S5. Sulfate evolution in reactor R1. Figure S6. sulfate evolution in reactor R0. Figure S7. Nitrogen compounds in reactor R1. Figure S8. Nitrogen compounds in reactor R0.

**Author Contributions:** Conceptualization, L.G. and A.B.; methodology, L.G., A.B. and J.R.; formal analysis and laboratory, J.R. and R.G.; investigation, J.R and A.B.; writing—original draft preparation, J.R. and A.B.; writing—review and editing, A.B., C.H. and R.B.; supervision, L.G.; funding acquisition, L.G. and A.B. All authors have read and agreed to the published version of the manuscript.

**Funding:** This research was funded by ANID—Government of Chile, Regular Fondecyt Project 1201258.

**Institutional Review Board Statement:** Not applicable.

**Informed Consent Statement:** Not applicable.

**Data Availability Statement:** Data supporting can be found emailing to: andrea.barahona@usm.cl.

**Acknowledgments:** The authors appreciate the collaboration of their friend Silvio Montalvo (RIP), who inspires them to continue researching.

**Conflicts of Interest:** The authors declare that the research was conducted in the absence of any commercial or financial relationships that could be construed as a potential conflict of interest.

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
