# Peer review of "Sequential Nitrification—Autotrophic Denitrification Using Sulfur as an Electron Donor and Chilean Zeolite as Microbial Support"

_water, doi:10.3390/w15010095_

Round 1

Reviewer 1 Report

This work describes an autotrophic denitrification process that uses sulfur as an electron donor and compares the operation of the reactor with and without zeolites as medium. Overall, there are some interesting findings in this manuscript, but the authors need to pay more attention to the logic and structure of the article to better convey the importance of this article to the readers.

Abstract

Line 20-21, Why the authors put this statement ‘Inhibition and accumulation of nitrite in the denitrification stage were achieved by increasing the concentration of nitrogen in the SWW’ in the abstract. From what follows in the manuscript, it appears that the accumulation of nitrites is not positive, but rather makes the overall efficiency worse. If the authors were trying to compare the two cases with and without zeolites, they should have at least included both cases in the abstract.

Introduction

This article mainly describes the sulfur-based autotrophic denitrification process, but there are few citations on this part in the introduction section. I suggest the authors add more on the coupling of the sulfur cycle and nitrogen cycle, especially to describe the mechanism, advantages and previous work clearly. Here list of some publications for the authors’ reference:

https://doi.org/10.1016/j.biortech.2018.06.062,

https://doi.org/10.1016/j.watres.2022.118404,

https://doi.org/10.1016/j.watres.2020.116619,

https://doi.org/10.1016/j.biortech.2020.123826,

Discussion

Line 255-262, from Figure 3, it can be seen that the concentration of sulfate reaches its maximum on day 11. Theoretically, more sulfate production means more nitrate was reduced, but from Figure 2, the accumulation of nitrate was still high on day 11. Why?

Line 342-351, In the abstract, the authors mentioned that zeolites can reduce 70% of the time in nitrification, but in the Results and Discussion section, the relevant description and explanation are not found.  Can the authors explain how they reached this conclusion?

Line 292, 328, 363, The VSS is mentioned several times in the manuscript, but the relevant data is not included in the manuscript and supplementary materials. Please provide?

Line 388-392, From Figure 8, the reduction of nitrate in the group with zeolites was not affected by the influent nitrogen load compared to the group without zeolites, can the authors further explain what the mechanism is?

Conclusion

Emphasize the contribution of this article to the field in the conclusion section, rather than simply summarizing what they have done.

Author Response

Dear reviewer, 
I have worked on your comments and improved on what you have requested, I hope these responses are satisfactory to what is expected.

The answers are in the attached document.

Best regards

Author Response

(The authors gave the same response as above.)

Reviewer 3 Report

This article reveals the importance of using sulfur as an electron donor and Chilean zeolite as microbial support for nitrogen removal. Some concerns need to be addressed:

1. The sulfur and Chilean zeolite were used independently or fabricate into compound?

2. Micro-morphology investigation, such as SEM of the filter materials could be provided.

3. What does that mean for ‘Inhibition and accumulation of nitrite…’ in line 20, it seems to present two opposite effects.

4. Line 91: for a batch mode, how to maintain a nitrogen loading rate (NLR) of 0.061 kg N/(m 3 ·d)?

5. The picture or schematic of aDN reactor is encouraged to provide in the text, e.g., incorporated in Fig. 1.

6. Since there is NH4Cl in table 1, how the authors prove that annamox process is present or not along with aDN.

7. For how long do the authors judge the completion of the microbial enrichment in Section 2.1 and 2.2 and is ready for the next inoculation. Please specify that.

8. How to exactly control the reactor temperature in Table 4?

9. Why Table 3 is placed behind Table 4?

10. Line 207-210, it really makes readers confused to grab the difference between these influents for DN and aDN. Why should SWW in Table 3 be added following the scenario in line 210?

11. Line 221: is it complete denitrification?

12. For each cycle in Fig. 2, a new batch of culture medium was added in the reactor? Please specify that.

13. By which means of did the N2 was determined in Fig. 5? Besides, there is no Y axis for Load changes.

14. Line 351: How to start each new cycle?

15. Line 386 and throughout the article: Replace ‘liquor’ with ‘liquid’ or ‘culture’.

Author Response

(The authors gave the same response as above.)

Round 2

Reviewer 2 Report

This acticle was finely modified as suggested, therefore I agreed for publication.